evolution, ecology, immunology

maternal antibodies, *Teladorsagia circumcincta*, immunoglobulin, Soay sheep, maternal effects, IgG

**Author for correspondence:**
Alexandra M. Sparks
e-mail: a.m.sparks@leeds.ac.uk

# Maternally derived anti-helminth antibodies predict offspring survival in a wild mammal

Alexandra M. Sparks[1,2,3], Adam D. Hayward[4], Kathryn Watt[1], Jill G. Pilkington[1], Josephine M. Pemberton[1], Susan E. Johnston[1], Tom N. McNeilly[4] and Daniel H. Nussey[1,2]

[1]Institute of Evolutionary Biology and [2]Institute of Immunology and Infection Research, School of Biological Sciences, University of Edinburgh, Edinburgh EH9 3FL, UK
[3]Faculty of Biological Sciences, School of Biology, University of Leeds, Leeds LS2 9JT, UK
[4]Moredun Research Institute, Pentlands Science Park, Bush Loan, Midlothian EH26 0PZ, UK

AMS, 0000-0002-7697-4632; ADH, 0000-0001-6953-7509; JMP, 0000-0002-0075-1504; SEJ, 0000-0002-5623-8902; TNM, 0000-0001-6469-0512; DHN, 0000-0002-9985-0317

The transfer of antibodies from mother to offspring provides crucial protection against infection to offspring during early life in humans and domestic and laboratory animals. However, few studies have tested the consequences of variation in maternal antibody transfer for offspring fitness in the wild. Further, separating the immunoprotective effects of antibodies from their association with nutritional resources provided by mothers is difficult. Here, we measured plasma levels of total and parasite-specific antibodies in neonatal (less than 10 days old) wild Soay sheep over 25 years to quantify variation in maternal antibody transfer and test its association with offspring survival. Maternal antibody transfer was predicted by maternal age and previous antibody responses, and was consistent within mothers across years. Neonatal total IgG antibody levels were positively related to early growth, suggesting they reflected nutritional transfer. Neonatal parasite-specific IgG levels positively predicted first-year survival, independent of lamb weight, total IgG levels and subsequent lamb parasite-specific antibody levels. This relationship was partly mediated via an indirect negative association with parasite burden. We show that among-female variation in maternal antibody transfer can have long-term effects on offspring growth, parasite burden and fitness in the wild, and is likely to impact naturally occurring host–parasite dynamics.

## 1. Introduction

Maternal effects can explain a considerable proportion of observed phenotypic variation in early-life traits and have important short- and long-term consequences for offspring fitness [1]. Mothers can influence their offspring's phenotype by varying the quality or quantity of nutritional resources and care provided pre- or post-natally [2,3]. Important non-nutritional resources are also transferred from mother to offspring including antioxidants, hormones and immunologically active molecules, via the egg or across the placenta and via milk [2]. The transfer of antibodies from mother to offspring can substantially impact offspring performance with consequences for evolutionary and disease dynamics in natural populations [2,4,5]. Maternal antibodies provide immune protection to offspring during a critical period when their immune system is not yet fully developed [6], and can result in longer-term priming of the offspring's developing immune response [7]. Failure of maternal antibody transfer is associated with high neonatal morbidity and mortality in domestic

livestock [8], and insufficient levels of maternal antibodies predict greater offspring susceptibility to infection and reduced offspring growth rates in a range of wild birds and mammals [9,10]. However, maternal antibodies may also have negative effects on offspring by suppressing the development of the humoural immune response [11]. Thus, while maternal antibody transfer is likely to be an important maternal effect in wild vertebrate systems, understanding its relationship with offspring fitness requires detailed data on offspring growth, immunity and infection status.

The impact of maternal antibodies on ecological, epidemiological and evolutionary dynamics in wild populations will depend critically on the amount—and causes—of variation in levels of antibody transfer among mothers and the consequences of this variation on maternal and offspring fitness [4,5]. The diversity and quantity of maternal antibodies transferred is expected to depend on the mother's nutritional state, her past exposure to parasites and her immune responses to these challenges [12]. These factors are likely to contribute to age-specific variation in maternal antibody transfer, with younger and older mothers generally having lower nutritional status and less effective immunity than prime-age mothers. Despite compelling evidence from laboratory, human and veterinary studies that maternal antibody transfer is closely related to maternal condition and important for offspring health and survival in early life, there remains limited understanding of the drivers of variation in maternal antibody levels and its impact on offspring fitness in the wild [2,4,5]. Measuring associations between phenotypes and fitness under natural conditions is by necessity observational rather than experimental. However, this provides insight into the operation of natural selection which cannot be obtained through experimental approaches. In correlational studies, a further challenge lies in identifying the mechanisms through which maternal antibodies influence offspring survival [5]. Transfer of parasite-specific antibodies to the offspring should provide early-life protection from infection, reducing parasite burdens and infection-related mortality risk [10]. Alternatively, maternal antibodies might provide immune protection, allowing the offspring to avoid using limited available resources on immunity, and invest more in growth and development, thus improving early survival prospects [5,13]. However, maternal antibody levels are likely to positively correlate with the quantity and quality of nutrition provided by the mother via the egg or during gestation and lactation [4]. Maternal antibody transfer could, therefore, positively predict offspring survival through covariation among maternal antibodies, maternal provisioning of resources to the offspring, and offspring body condition and growth rates, independent of any direct immune protection provided by the antibodies.

In ruminants, there is no prenatal transfer of immunoglobulins from the mother to the fetus across the placenta, and consequently offspring consumption of colostrum has a fundamental role in maternal antibody transfer [6]. The non-specific pinocytotic absorption of immunoglobulins from the intestine of ruminants diminishes rapidly after birth and ceases by 24–36 h post-partum due to gut closure [14,15]. After gut closure, immunoglobulins in milk continue to provide local protection against enteric infection [15]. As neonatal ruminants do not synthesize their own immunoglobulins for several weeks, immunoglobulin levels in the blood are entirely of maternal origin [16] and reflect a readily measurable indicator of maternal antibody transfer. The colostrum and milk also provide essential nutrition for neonate growth and survival in mammals [15]. In ruminant colostrum, immunoglobulins account for 70–80% of total protein [17] with IgG being the predominant immunoglobulin (65–90% of total antibodies [15]). Measuring total IgG levels in neonatal blood thus provides a useful proxy for total antibody and protein transferred by the mother during early life, and potentially allows the separation of the beneficial effects of parasite-specific antibodies from the general quality of the colostrum [4].

The objective of this study was to examine the causes and consequences of variation in maternal antibody transfer in a natural population with a well-understood host–parasite system. To achieve this, we measured maternally transferred antibody levels in blood samples taken from lambs within 10 days of birth (henceforth 'neonatal antibodies') in a wild population of Soay sheep (*Ovis aris*) over a 25-year study period. In this population, infection with strongyle nematodes has well-documented negative effects on health and survival, particularly for lambs [18–21]. Circulating levels of anti-strongyle antibodies in summer predict both summer strongyle burden and subsequent over-winter survival in adults [22]. First, we aimed to determine the causes of variation in maternally transferred antibody levels, by investigating which offspring and maternal characteristics were associated with neonatal parasite-specific antibody and total IgG levels. Next, we aimed to determine the fitness benefits of maternal antibodies and dissect whether any effects were driven by parasite-specific antibodies (enhanced immunity to worms) or total IgG levels (enhanced maternal resource provisioning in colostrum). We achieved this by testing whether neonatal antibody levels predicted offspring body weight, parasite burden and antibody levels at weaning (four months old), as well as survival to weaning and first-winter survival. Having found that neonatal anti-strongyle IgG antibody levels positively predicted offspring survival, we addressed whether this relationship was a direct association, or whether and how it was mediated through indirect associations with neonatal total IgG levels, offspring weight and parasite burden. To accomplish this, we used a structural equation modelling (SEM) approach. Overall, our results demonstrate how variation among females in maternal antibody transfer can have important long-term consequences for offspring parasite resistance, growth and survival in the wild.

## 2. Methods

### (a) Study population and laboratory assays

The Soay sheep is an ancient breed of domestic sheep that has been living under unmanaged conditions on the remote St Kilda archipelago for several millennia. Sheep living in the Village Bay area of the island of Hirta have been the subject of a long-term study since 1985 [23]. In April each year, around 95% of all lambs born in this area are caught and individually tagged, weighed and blood-sampled, at a mean capture age of 3 days (s.d.: 2.7 days, 1990–2015 data). Each August, around 50% of the study population is re-captured, weighed and have blood and faecal samples collected. Whole blood samples are collected into heparin tubes, centrifuged at 3000 r.p.m. for 10 min, and plasma removed and stored at −20°C. August strongyle faecal egg count (FEC) is estimated from faecal samples as the number of eggs per gram using a modified McMaster technique [24]. Three species contribute the majority of strongyle FEC: *Teladorsagia circumcincta*, *Trichostrongylus axei* and *Trichostrongylus vitrinus* [20]. These parasites have a

direct life cycle: eggs are shed in the faeces and develop into larvae on pasture, which are then consumed by sheep and migrate and mature into adults in the guts of their hosts [19]. Although lambs are not weaned until around four months of age, exposure to strongyle larvae may occur at a very young age: lambs have been observed to nibble grass within their first week of life and continue to eat grass while suckling [23,25]. Most mortality occurs over winter, and it is highly variable between years due to variation in density, weather and the proportion of vulnerable individuals [26]. Regular censuses and mortality searches during the winter months result in most carcasses being discovered and provide accurate death date information.

Levels of IgA, IgG and IgE antibodies binding to antigens of the third larval stage of *T. circumcincta* (henceforth, 'anti-Tc antibodies') and total IgG levels were measured using ELISA from blood samples obtained from neonatal lambs during lambing seasons between 1990 and 2015 (see electronic supplementary material, Methods for full laboratory details developed from [22] and sample selection criteria). Our previous work suggested that anti-Tc antibodies represent a cross-reactive antibody response which bind antigen from a wide range of nematode species [27]. A description of the different antibody classes (isotypes) in mammals is available in the electronic supplementary material. There was no evidence of sample degradation in our dataset (see electronic supplementary material, figure S2). Our final dataset comprised neonatal antibody measures (anti-Tc IgA, IgE, IgG and total IgG) for 3379 lambs caught within 10 days of birth from 845 mothers. Distributions of each neonatal antibody measure are provided in electronic supplementary material, figure S3. We calculated 'neonatal survival' and 'first winter survival' for lambs as survival to 1 August in the birth year and to 1 May in the subsequent year, respectively, using death date and census information. Only 5% of lambs did not survive to August, while 55% did not survive their first winter. Such low levels of neonatal mortality in our dataset, despite high neonatal mortality in general (24% of all mortalities in the study area [28]), is likely due to the fact that many lambs that die neonatally perish in the first few days after birth, before they can be captured and sampled.

## (b) Statistical analyses
### (i) Models of neonatal antibody levels
We examined potential causes of variation in neonatal anti-Tc IgA, IgE, IgG and total IgG levels using linear mixed effects models (LMMs) in the package lme4 v. 1.1-21 [29] in R 3.5.3 [30]. Models included antibody levels as a response variable with maternal identity, year and ELISA plate as random effects to account for variation among mothers, years and laboratory conditions. Fixed effects included lamb sex (factor), twin status (factor), capture age of the lamb (in days), birth weight of the lamb and maternal age (linear and quadratic terms), as these terms had previously been identified as important predictors of lamb state and fitness [31–33]. Since Soay lambs are not caught at birth and weight is expected to vary with age after birth, birth weight was estimated by taking the residuals from a linear model of capture weight with capture age as a factor. Initially, we compared neonatal antibody models including linear or quadratic functions of capture age with threshold functions of age with a single threshold which we varied from days 1 to 9 [34]. The best model of capture age was chosen based on the lowest AIC value, unless the difference between the two best fitting models was less than two AIC values, in which case the most parsimonious model was chosen. Models were then simplified by stepwise deletion, sequentially removing fixed effects with the lowest $t$-values, and statistical significance was determined using likelihood ratio tests until a base model containing only significant ($p < 0.05$) fixed effects was left. All dropped non-significant terms were then added back into the

base model one at a time and significance tested using the same criteria. From the base model of each antibody measure, we finally tested a fixed effect of the mother's plasma anti-Tc antibody levels of the same isotype in the August prior to birth. In ruminant colostrum, the majority of immunoglobulins are derived by transfer from the bloodstream [15]. Previously, it was shown that plasma anti-Tc IgA, IgE and IgG levels measured in August over 26 years are very repeatable in adult Soay sheep, with high correlations between measures in consecutive years [35]. As a result, we predicted that antibody levels in the previous August may reflect those in April of the following year. As not all mothers were sampled the previous summer, sample sizes were reduced for these models (anti-Tc IgA = 1774/3213, anti-Tc IgE = 1779/3218, anti-Tc IgG = 1765/3212, total IgG = 1764/3214).

## (ii) Models of offspring growth, parasite burden and survival
To test whether neonatal antibody levels predicted subsequent lamb weight, parasite burden and antibody levels at four months old, as well as their neonatal and first-winter survival, we built generalized LMMs of each of these measures. Details of the fixed effects and random effects included in each model are provided in electronic supplementary material, table S1 and follow previous work [22]. For the August strongyle FEC model, a range of model fits were compared with and without zero-inflation in the package glmmTMB v. 1.0.1 [36] and the model with lowest AIC chosen. For the analyses of survival, continuous variables in the fixed effects structure were rescaled to mean 0 and standard deviation 1 to help model convergence. Due to the strong negative association between neonatal anti-Tc antibodies and lamb capture age, neonatal antibody levels were corrected for capture age by taking residuals from a model with thresholds at day 4 for anti-Tc IgA and IgE and day 6 for anti-Tc IgG or a linear relationship for total IgG (see Results). Models were not simplified prior to inclusion of neonatal antibody levels. To each of the models (electronic supplementary material, table S1), we added each neonatal antibody measure separately and tested the significance of these terms using likelihood ratio tests.

## (iii) Structural equation modelling
Our GLMMs revealed a positive association between neonatal anti-Tc IgG and first-winter survival in Soay lambs (see Results). In order to further determine the extent to which neonatal anti-Tc IgG levels directly influenced first-winter survival, and the extent to which this association was indirectly mediated by associations with August weight, August strongyle FEC or August anti-Tc IgA levels, we used SEMs. We included links based on significant associations from previous work and from the current study (for full details, see electronic supplementary material, table S2) to produce an *a priori* diagram (figure 1a) based on evidence for likely causal relationships [37]. LMMs found a negative association between neonatal anti-Tc IgG and both August anti-Tc IgA and IgG levels (electronic supplementary material, table S5). Previous work showed that lamb August anti-Tc IgA was the best predictor of August FEC [22], so we included this measure in our SEMs to capture lamb immune resistance to worm infection at weaning. We also ran the SEM with August anti-Tc IgG instead and this produced similar results (see electronic supplementary material, figure S8a,b) which is unsurprising given that these antibodies are positively correlated [22]. We were able to draw the starting path diagram assuming causality because most of our variables are separated in time (April, August and April in the subsequent year, figure 1a). Bidirectional relationships were fitted between the measures in April (neonatal antibodies and birth weight) as causality here was unclear. Our initial model assumed directional relationships among August measures: anti-Tc IgA and weight influenced FEC (figure 1a; electronic supplementary material, table S2). This allowed us to test whether first-winter survival was directly

Proc. R. Soc. B 287: 20201931

**4**

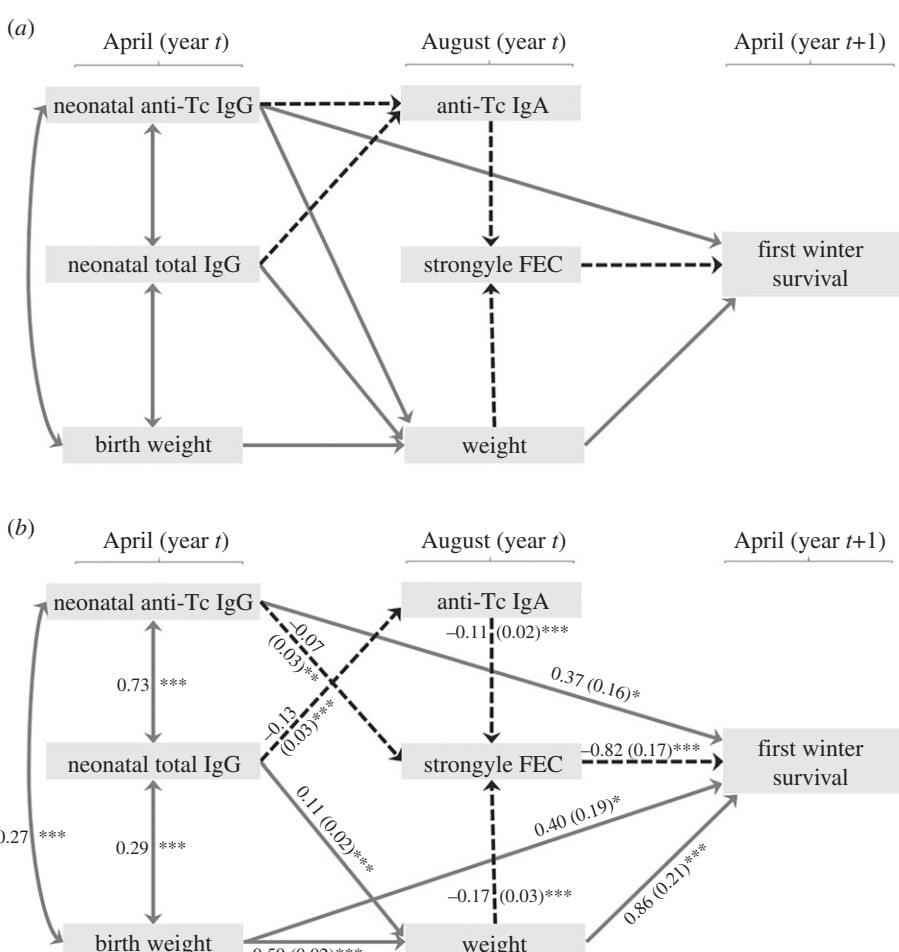

**Figure 1.** (a) A priori SEM linking neonatal anti-*T. circumcincta* and total IgG antibodies to first-winter survival based on associations in this study and from previous work on this system (see electronic supplementary material, table S2 for details). (b) Final SEM with values on arrows indicating standardized path coefficients with standard errors in brackets, except for the bidirectional links between neonatal measures which show correlation coefficients. Missing paths in the a priori model were added where indicated using Shipley's test of d-separation and unsupported paths were removed based on p-values ≥0.05. Effects are separated into positive (grey full lines) and negative (black dashed lines) effects with p-values indicated by asterisks (***$p < 0.001$, **$p < 0.01$ and *$p < 0.05$).

explained by each August measure or by indirect effects of either weight via FEC or antibody levels via FEC (figure 1a).

All models included random effects of year and maternal identity. We corrected the neonatal antibodies for the best fitting capture age function from our previous models (see Results) and birth weight for capture age as a factor. All continuous variables were divided by 2 s.d., while categorical variables (i.e. survival) were unstandardized, generating standardized coefficients that were comparable [38,39]. August strongyle FEC (plus the lowest non-zero FEC value of 50) was natural log-transformed to adhere to assumptions of residual normality. Our model was based on 1389 lambs which had data on neonatal antibody levels, August measures and first-winter survival. We first ran separate models, and then combined these models using piece-wiseSEM v. 2.1.0 [40]. Shipley's test of d-separation was used to determine if any paths were missing, which were sequentially added. Unsupported paths were removed based on p-values ≥ 0.05. Standardized path estimates linking neonatal anti-Tc or total IgG and survival were determined as the products of standardized coefficients along each path.

## 3. Results

### (a) Models of neonatal antibody levels

The four neonatal antibody measures (measured within 10 days of birth) were positively correlated with one another

(electronic supplementary material, figure S4, $r = 0.323$–0.776), with the strongest correlation between anti-Tc IgG and total IgG levels. Maternal effects explained 49%, 61%, 39% and 14% of the phenotypic variance in anti-Tc IgA, IgE and IgG and total IgG levels, respectively (electronic supplementary material, table S3). Capture age, birth weight, twin status and sex were associated with neonatal antibody levels (electronic supplementary material, table S3). The best fitting function of capture age on neonatal antibody levels was defined by a threshold model with an inflection point at day 4 for anti-Tc IgA and IgE, and day 6 for anti-Tc IgG (ΔAIC to next best model for IgA = 16.496, IgE = 0.755, IgG = 1.197; electronic supplementary material, table S4). Anti-Tc IgA and IgE antibody levels declined with capture age, with a steeper linear decline between 0 and 4 days old (electronic supplementary material, table S3 and figure S5a,b). Anti-Tc IgG levels declined with capture age up to 6 days after which there was no significant linear relationship with capture age, while total IgG levels declined linearly without a detectable threshold (electronic supplementary material, table S3 and figure S5c,d). Birth weight was positively associated with all neonatal antibody levels (electronic supplementary material, table S3 and figure S6). Twins were more likely to have higher levels of all antibodies, while males tended to have lower anti-Tc IgG and total IgG levels (electronic supplementary material, table S3). There was a quadratic association of

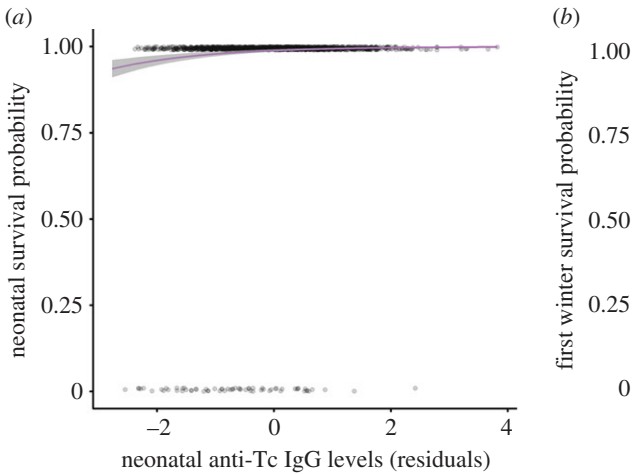
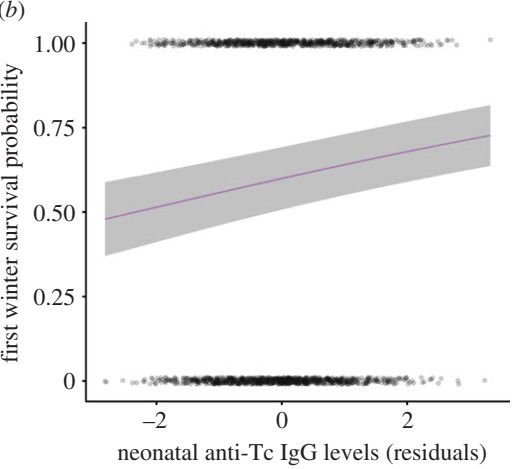

**Figure 2.** Associations between neonatal anti-*T. circumcincta* IgG levels (corrected for capture age and standardized, see Methods for full details) and neonatal survival (*a*) and neonatal anti-*T. circumcincta* IgG levels and first-winter survival (*b*) in Soay sheep. Plots show raw data with GLMM predictions and standard errors estimated for singleton female lambs with average values for all continuous fixed effects in the model specified in electronic supplementary material, table S7 with just neonatal anti-*T. circumcincta* IgG levels included.

maternal age with neonatal anti-Tc IgE and IgG and total IgG levels, in which offspring of the youngest and eldest mothers had lower antibody levels (electronic supplementary material, table S3 and figure S5*f*–*h*). By contrast, neonatal anti-Tc IgA levels declined linearly with maternal age (electronic supplementary material, table S3 and figure S5*e*), but a similar quadratic association was seen if birth weight was dropped from the model (maternal age$^2$ – $\chi_1^2 = 13.407$, $p < 0.001$). All four neonatal antibody levels were significantly and positively associated with the mother's plasma antibody levels of the same isotype measured in the preceding August (electronic supplementary material, table S3 and figure S7).

## (b) Models of offspring growth, parasite burden and survival

Neonatal antibody levels were generally negatively associated with lamb antibody levels measured at around four months old (August) when most lambs were nearly weaned (electronic supplementary material, table S5). These negative associations were significant for August anti-Tc IgG and IgA, but not for IgE (electronic supplementary material, table S5). There were significant positive associations between all four neonatal antibody levels and August weight, independent of birth weight (electronic supplementary material, table S6). Generally, we found limited evidence that neonatal antibody levels significantly predicted August strongyle FEC; relationships were in the predicted negative direction for anti-Tc IgA and anti-Tc IgG, but not for anti-Tc IgE or total IgG (electronic supplementary material, table S6).

There were positive associations between all four neonatal antibody levels and survival to four months (electronic supplementary material, table S7). However, neonatal antibody levels are correlated, and so we tested independent associations between antibody levels and survival by including all four antibody measures in the same model. In this model, only neonatal anti-Tc IgG levels predicted survival independently (anti-Tc IgA: $\beta = 0.136 \pm 0.135$ s.e., $\chi_1^2 = 1.039$, $p = 0.308$; anti-Tc IgE: $\beta = -0.158 \pm 0.150$ s.e., $\chi_1^2 = 1.077$, $p = 0.299$; anti-Tc IgG: $\beta = 0.552 \pm 0.218$ s.e., $\chi_1^2 = 6.747$, $p = 0.009$; Total IgG: $\beta = 0.108 \pm 0.178$ s.e., $\chi_1^2 = 0.369$, $p = 0.544$, figure 2*a*). Of the four antibody measures, only neonatal anti-Tc IgG levels

significantly predicted first-winter survival; lambs with higher anti-Tc IgG levels were more likely to survive to the following spring independent of August weight (electronic supplementary material, table S7; figure 2*b*).

## (c) Structural equation models

SEMs based on the *a priori* path diagram in figure 1*a* identified two missing paths which were included in our final model: between neonatal anti-Tc IgG and August FEC ($\beta = -0.070 \pm 0.026$ s.e., *Fisher's* $C = 7.438$, $p = 0.007$), and a direct path between birth weight and survival ($\beta = 0.403 \pm 0.195$ s.e., *Fisher's* $C = 2.065$, $p = 0.039$). Two paths in the model in figure 1*a* were not supported and were removed from the model: the paths linking neonatal anti-Tc IgG with August anti-Tc IgA ($\beta = -0.021 \pm 0.042$ s.e., *Fisher's* $C = 0.248$, $p = 0.619$) and neonatal anti-Tc IgG with August weight ($\beta = 0.058 \pm 0.031$ s.e., *Fisher's* $C = 3.446$, $p = 0.064$). The final SEM, presented in figure 1*b*, adequately fitted the data and no missing paths were expected (*Fisher's* $C = 19.163$, $p = 0.260$). SEMs including August anti-Tc IgG instead of IgA produced qualitatively similar results, although August anti-Tc IgG levels were negatively associated with neonatal anti-Tc IgG rather than with neonatal total IgG, and neonatal total IgG was weakly positively related to August FEC in that model (see electronic supplementary material, figure S8*a*,*b*). Here, we focus on results from the model including August anti-Tc IgA.

Higher levels of neonatal anti-Tc IgG were associated with higher first-winter survival via both a direct path and an indirect path mediated by August FEC (figure 1*b* and table 1). The strongest effects on survival were direct effects of August weight and FEC, with the direct, independent effects of birth weight and neonatal anti-Tc IgG on survival being around half as strong and comparable to each other (figure 1*b*). Neonatal anti-Tc IgG levels also positively predicted survival via an indirect effect on August FEC, with higher antibody levels associated with reduced strongyle FEC in summer which in turn predicted higher survival (figure 1*b* and table 1). The estimated effect size associated with this indirect path was only around a sixth of the magnitude of the direct path between neonatal anti-Tc IgG and survival (table 1). The positive relationship between neonatal

**Table 1.** Standardized path estimates showing the total influence of each of the neonatal variables (anti-Tc IgG, total IgG and birth weight) on over-winter survival of Soay lambs from a SEM (figure 1*b*). Where paths include more than one effect, standardized path estimates were calculated as the product of the composite paths.

| path | standardized path estimate |
| --- | --- |
| *neonatal anti-Tc IgG* | |
| neonatal anti-Tc IgG → survival | 0.371 |
| neonatal anti-Tc IgG → August strongyle FEC → survival | 0.057 |
| *neonatal total IgG* | |
| neonatal total IgG → August weight → survival | 0.091 |
| neonatal total IgG → August weight → August strongyle FEC → Survival | 0.015 |
| neonatal total IgG → August anti-Tc IgA → August strongyle FEC → survival | −0.012 |
| *birth weight* | |
| birth weight → survival | 0.403 |
| birth weight → August weight → survival | 0.504 |
| birth weight → August weight → August strongyle FEC → survival | 0.082 |

antibodies and survival was not mediated by any increase in offspring antibody levels in summer: neonatal total IgG levels were actually negatively associated with August anti-Tc IgA levels and the path from neonatal total IgG to August anti-Tc IgA to FEC to survival produced an overall very weak negative effect (table 1 and figure 1*b*). Neonatal total IgG levels were also weakly positively associated with survival indirectly via August weight (table 1 and figure 1*b*). Therefore, neonatal total IgG levels had both positive indirect effects on survival via higher August weight and negative indirect effects on survival via lower August anti-Tc IgA levels and subsequent higher August FEC. However, the combined indirect effects of neonatal total IgG levels on first-winter survival were positive (table 1).

## 4. Discussion

In this study, we aimed to determine the causes and consequences of variation in maternally transferred antibody levels for offspring fitness in a natural population using 25 years of measurements of neonatal antibodies in wild Soay sheep. For causes of variation, maternal identity explained a considerable proportion of observed variation in offspring neonatal parasite-specific antibody levels. Whether this is due to consistent levels of antibodies transferred to offspring or consistent inter-year immune responses of females is unknown. Further, variation in maternal antibody transfer is related to maternal age and previous parasite-specific antibody levels in the mother's plasma, suggesting that maternal antibody transfer is related to maternal condition and the mother's ability to produce effective immune responses. In terms of the consequences of maternal antibodies, we demonstrate that among-female variation in maternal transfer of parasite-specific antibodies is under natural selection via both offspring neonatal and first-winter survival. This represents rare evidence that variation in maternal antibodies against a naturally occurring parasitic helminth has downstream consequences for offspring fitness in the wild. We aimed to dissect whether there was a direct or indirect association between maternal antibodies and fitness. We found that the association was not mediated by

covariation with neonatal total IgG (a proxy for colostrum quality and quantity), offspring birth weight or growth rates. Instead, we find support for offspring fitness benefits of increased maternal transfer of parasite-specific antibodies via reduced parasite burdens after weaning and for an unexplained direct association with over-winter survival. Our results provide evidence that maternal antibodies can have long-lasting effects on offspring phenotype, parasite burden and fitness in the wild, and thus that among-female variation in maternal transfer of immunity could impact the population, evolutionary and epidemiological dynamics of natural systems.

Consistent with previous studies, neonatal antibody levels were related to maternal age and the mother's own previous immune phenotype. As expected, we found that neonatal antibody levels tended to be lower in offspring of the youngest and eldest mothers. Such quadratic maternal age effects are widely observed in wild vertebrates [41–43], and the pattern mirrors trends with age in survival, reproductive performance, weight and parasite burden in the Soay sheep [44]. Maternal age is a strong and widespread predictor of maternal condition and performance in studies of wild vertebrates [45], and it seems likely that the effect of maternal age on maternal antibody transfer in Soay sheep reflects reduced condition or immune development and function in the youngest and eldest females. Further support for this is provided by the positive association between neonatal antibody levels and the mother's own anti-strongyle antibody levels in the summer prior to giving birth. Previous experimental studies in captive rodents have demonstrated that maternal antibodies provide strain-specific protection which depends on the infection history of the mother [46,47]. Given that exposure to worms starts early, we would expect that all females would be sufficiently exposed to have developed an acquired immune response to strongyles by sexual maturity, despite exposure varying in time and space [19]. Given the high repeatability of antibody levels in adults, and positive associations with fitness and negative associations with FEC, we believe that these antibody levels reflect variation in the immune response rather than just differences in exposure in this population [22,35]. The substantial, consistent differences among mothers in neonatal antibody levels of their

offspring (indicated by the large maternal effect) suggest maternal genotype and prior maternal environment may play an important role in driving the variation in maternal antibody transfer we observe in wild Soay sheep. Further study using available genetic and environmental data for this system could help elucidate their relative importance.

We found that neonatal antibody levels predicted lamb growth rates in wild Soay sheep. Increased maternal antibodies have been associated with improved growth rates of offspring in birds [9,13] and laboratory mice [48], and we provide evidence that maternal antibodies positively predict lamb growth over the weaning period in our models. The simplest explanation for this relationship is that maternal antibody levels correlate with the quality of nutrition provided by the mother to the offspring which in turn predicts offspring growth. Previous studies of domestic ruminants have focused on neonatal serum total IgG and found positive associations with growth and survival of offspring [8]. We found that total IgG levels in neonates, but not parasite-specific IgG levels, predicted body mass at four months old (figure 1b), suggesting an important role for high milk protein content and quantity, rather than positive immune-mediated effects of parasite-specific antibodies, for offspring growth [5,13].

Neonatal parasite-specific IgG levels positively predicted both lamb survival to four months ('neonatal survival'; figure 2a) and first-winter survival (figure 2b). Mortality during these two periods is likely due to different causes and may thus reflect different selective pressures on maternal antibody transfer in the wild. Neonatal mortality occurs when lambs are highly dependent on their mothers for nutrition and, therefore, dependent on maternal condition and investment in development and lactation [23]. On St Kilda, lambs are exposed to strongyle parasites very early: they begin eating grass within their first few weeks and may be hosts to egg-producing adult worms by 45 days of age [23]. Although all four neonatal antibody measures significantly predicted neonatal survival, when fitted together in the same model only the anti-Tc IgG effect remained significant, implying that it best explains the association with survival across these correlated measures. This association is independent of the well-documented relationship between birth weight and neonatal survival [23] and suggests a potential role for protective maternal transfer of immunity. Although neonatal antibody levels in plasma declined over the first 10 days of life consistent with gut closure (electronic supplementary material, figure S5a–d), mothers continue to deliver antibodies to the offspring's gut via milk, and these maternal antibodies have a role in local protection against enteric infection during lactation [15]. Our results suggest that neonatal parasite-specific antibodies, measured within the first 10 days of life, may provide an indicator of consistent differences among mothers in the delivery of maternal antibodies across the entire lactation period. However, further work is required to determine whether strongyle infection and damage during early life could play a role in neonatal mortality, and whether and how maternal transfer of strongyle-specific antibodies in the colostrum influences any early strongyle-induced pathology.

Neonatal anti-Tc IgG levels were the only antibody measure to significantly predict lamb first-winter survival. The association was independent of neonatal total IgG and birth weight, as well as weight at four months old, showing that maternal antibodies are important over and above total protein received by the neonate and improvements to growth and condition. Lambs are weaned and largely independent of their mothers by the onset of their first winter, and mortality during this period is likely to reflect their ability to survive the interacting pressures of food limitation, thermoregulatory challenges and parasites across the winter. If the association between neonatal anti-Tc IgG and winter survival was driven by maternal transfer of immunity, we would expect that it would largely manifest via reductions in August FEC. SEMs did detect a weak path via August FEC, but this explained only a small proportion of the direct relationship between neonatal anti-Tc IgG and winter survival (table 1).

Previous studies on protective immunity to T. circumcincta infection have mainly focused on the importance of IgA and IgE. In lambs, reduced FEC is associated with increased IgA levels directed at worm growth, fecundity and the inhibition of larvae development [49,50]. In resistant sheep, a hypersensitivity response results in the expulsion of incoming larvae, and the arrestment of larvae development moderated partly by IgA [51]. IgE levels and IgE-dependent mast cell degranulation are also negatively associated with FEC and worm burden [52,53]. While IgA and IgE are produced more locally at the mucosa, IgG is the dominant antibody isotype in circulation [15]. Studies investigating the role of IgG in T. circumcincta infection have documented negative associations between parasite-specific IgG and FEC or worm burden [54,55], while IgG has been implicated indirectly by protective effects of complement in other ruminant–worm interactions [56,57]. In laboratory mice, IgG has been documented to have an important role in passive immunity, and mediating protective immunity, to several nematode species [58–61]. Previously, using samples collected in August, we found that circulating levels of anti-Tc IgG, but not IgA or IgE, predict over-winter survival in adults, but there was no association between any of these antibodies and over-winter survival in lambs [22]. We have also shown that high plasma anti-Tc IgG is associated with high local anti-Tc IgG and IgA [62]. Our results provide further evidence of the importance of the IgG isotype in resistance to strongyle infection in this system, via associations between maternally derived levels of anti-Tc IgG and offspring survival.

How neonatal anti-Tc IgG levels predict first-winter survival is unclear. Interestingly, higher maternal antibody levels in lambs do not predict increased parasite-specific antibody levels at four months old. Instead most relationships are negative, consistent with vaccination studies in humans and birds in which maternal antibodies suppress offspring humoural responses to vaccines [11,63]. This implies that any protective benefit of maternal antibodies in our study system is not via improvements in lamb immune development. One possibility is that our measure of parasite burden, FEC, is not sensitive or specific enough to detect the impacts of maternal antibodies on strongyle infection that matter for offspring mortality. Our strongyle FEC measure includes five different species, with different sites of infection, age-specificity and pathogenicity [19,20]. Maternal antibodies may impact which species are able to establish in the lamb's gut and result in a less pathogenic nematode community, but an identical FEC. Furthermore, studies in domestic sheep suggest that the inflammatory and hypersensitive immune response produced by lambs in response to strongyle infection result in appetite

loss and diarrhoea [51], and maternal antibodies may provide early shielding from infection which limits such damaging immune responses, with positive effects on subsequent lamb health that are not detectable from subsequent weight, FEC or antibody measures. While speculative, these ideas could be investigated further with more intensive sampling during early life and using non-invasive meta-barcoding techniques to unveil the development of nematode community structure within the host [64]. Overall, these results provide important new support for an impact of maternal antibodies on offspring fitness in the wild, while also highlighting how potentially subtle and complex the effects of maternal antibodies on offspring health and fitness may be under natural infection.

Ethics. All animal work has been carried out according to UK Home Office procedures and is licensed under the UK Animals (Scientific Procedures) Act of 1986 (licence no. PPL60/4211).

Data accessibility. Data are available from the Dryad Digital Repository: https://dx.doi.org/10.5061/dryad.ttdz08kvx [65].

Authors' contributions. This study was conceived by A.M.S., D.H.N., T.N.M., S.E.J. and developed by A.D.H. J.M.P., J.G.P. and D.H.N. manage the long-term Soay sheep study system. Samples were collected by J.M.P. and J.G.P. Laboratory work was undertaken by K.W. and A.M.S. with some reagents provided by T.N.M. A.M.S. performed the statistical analyses with inputs from D.H.N. and A.D.H. A.M.S. wrote the first draft of the manuscript with input from D.H.N. and A.D.H. All authors provided comments on the manuscript and gave final approval for publication.

Competing interests. We declare we have no competing interests.

Funding. The Soay sheep project is supported by grants from the Natural Environment Research Council. D.H.N. was supported by a Biotechnology and Biological Sciences Research Council David Phillips fellowship (BB/H021868/1), A.D.H. was supported by a Moredun Foundation Fellowship and A.M.S. was supported by a Medical Research Council PhD studentship (award 1369297). T.N.M. is supported by the Scottish Government Rural Affairs, Food and the Environment (RAFE) Strategic Research Portfolio 2016–2021 and S.E.J. is supported by a Royal Society University Research Fellowship (UF150448).

Acknowledgements. We thank National Trust for Scotland, Scottish Natural Heritage, QinetiQ, Eurest and Kilda Cruises for logistical support on St Kilda. We further thank the many field volunteers and all those who have contributed to funding the Soay sheep project since its inception, in particular Tim Clutton-Brock. We would also like to express our gratitude to Ian Stevenson of Sunadal Data Solutions for long-term support of data collection and data management.

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
