## [Reviewer comments · Proceedings of the Royal Society B: Biological Sciences]

Review History

RSPB-2020-1931.R0 (Original submission)

Review form: Reviewer 1

Recommendation

Accept with minor revision (please list in comments)

Scientific importance: Is the manuscript an original and important contribution to its field?

Excellent

General interest: Is the paper of sufficient general interest?

Good

Quality of the paper: Is the overall quality of the paper suitable?

Excellent

Is the length of the paper justified?

Yes

Should the paper be seen by a specialist statistical reviewer?

No

Do you have any concerns about statistical analyses in this paper? If so, please specify them explicitly in your report.

No

It is a condition of publication that authors make their supporting data, code and materials available - either as supplementary material or hosted in an external repository. Please rate, if applicable, the supporting data on the following criteria.

Is it accessible?

Yes

Is it clear?

Yes

Is it adequate?

Yes

Do you have any ethical concerns with this paper?

No

Comments to the Author

In this manuscript, the authors utilize a 25-year dataset from Soay sheep naturally infected with a strongyle parasite to test if maternally-derived antibodies impact later parasite burdens, offspring size and first winter survival. The authors measured parasite-specific levels of IgG, IgE and IgA, and total IgG levels. Total IgG levels predicted offspring growth and was interpreted as a measure of nutritional contributions from mothers to offspring. They found a positive association between levels of all four antibody measures and survival to four months; however, parasite-specific IgG levels predicted lamb survival independently. Furthermore, this relationship was independent of lamb body weight. There was also an indirect effect of parasite-specific IgG levels on lamb survival. Lambs with higher parasite-specific IgG shortly after birth had lower fecal egg counts of the strongyle parasite in the summer and lower fecal egg counts predicted higher survival. This is a compelling demonstration from a natural population that variation among females in maternal antibody transfer can have important consequences for offspring parasite resistance, growth, and survival.

General Comments:

1. The critical statements in the Introduction and Discussion that should frame what the authors wanted to learn from the study should be re-written to increase clarity and impact by following the structure "to learn x, we did y". For example, the last paragraph of the Introduction beginning at line 88 (and most of this paragraph) is a statement of "we did y" but does not clearly communicate what the authors wanted to learn or what knowledge gap they wanted to fill by using these samples and analyses. Similarly, the first paragraph of the Discussion beginning at line 295 is a statement of what was done, rather than what was learned. In both instances, the critical information you want to communicate about what you hoped to learn from the study is buried later in the paragraphs. These paragraphs would be more impactful if the statements of what you wanted to learn were placed at the beginning before describing the techniques.
2. When were the plasma samples analysed to measure antibody levels relative to sample collection? Were all of the samples assayed at one time or were they assayed as they were collected? If the samples were all analysed at the same time, then antibody levels from the oldest samples would be expected to be lower simply due to sample degradation over the 25-year period. It is not entirely clear but it seems all samples were assayed at the same time based on this sentence (line 26-27 supplementary info), "In order to minimise confounding capture year effects with plate to plate variation, each plate included samples from two years paired at random." It is critical to account for potential effects of antibody degradation in storage due to sample age or at least demonstrate that sample storage time was unrelated to the measured antibody levels.

3. Given that IgA is primarily associated with the gut and IgE is the primary mediator of the parasitic immune response, it was surprising that parasite-specific IgG levels were the strongest predictor of lamb survival in the first year. Through what potential mechanisms do you think IgG provides protection in this system that would be unique and distinct from IgA and IgE?

Specific Comments:

1. The Abstract could be strengthened to better frame the knowledge gap. For example, the first sentence tells readers that maternal antibodies are crucial for survival and then, the second sentence tells readers that we don't really know if maternal antibodies are critical for fitness.
2. Line 107: To people unfamiliar with this population, individuals living in the Village might be assumed to refer to human individuals.
3. Line 126: What do you mean by "IgG, IgA and IgE activity against antigens"?
4. Lines 314-315: A reminder of how maternal condition was quantified would be helpful.

Review form: Reviewer 2

Recommendation

Accept with minor revision (please list in comments)

Scientific importance: Is the manuscript an original and important contribution to its field?

Excellent

General interest: Is the paper of sufficient general interest?

Excellent

Quality of the paper: Is the overall quality of the paper suitable?

Excellent

Is the length of the paper justified?

Yes

Should the paper be seen by a specialist statistical reviewer?

No

Do you have any concerns about statistical analyses in this paper? If so, please specify them explicitly in your report.

No

It is a condition of publication that authors make their supporting data, code and materials available - either as supplementary material or hosted in an external repository. Please rate, if applicable, the supporting data on the following criteria.

Is it accessible?

Yes

Is it clear?

Yes

Is it adequate?

Yes

Do you have any ethical concerns with this paper?

No

Comments to the Author

This manuscript presents a well-conducted correlative study on an important topic, the potential effect of maternal antibodies in natural settings. It is based on a unique data set and the findings are clear, highlighting the potential role of maternally transferred immunoglobulins on the protection of lambs against naturally occurring endoparasites to which they are exposed after birth in a wild living sheep population.

In general, the manuscript is well written, the analyses are sound and the figures and tables provide useful results. It has nevertheless some weaknesses that require some consideration:

(1) The fact that the study is not experimental is a weakness that needs to be acknowledged, and it is important that readers are made aware that earlier experimental studies have been carried out on the topic, even if for most of those it was done using relatively artificial settings involving captive rodents (e.g., Staszewski et al. 2012, Gomez-Chamoro et al. 2019).

(2) The potential role of the different types of immunoglobulin isotypes, IgA, IgE and IgG, is not discussed and specifically investigated. Even if not much information is available from the literature, could this be outlined more clearly? (see other comments below)

(3) When considering the role of antibodies in epidemiology, ecology and the evolution of immune defenses, the specificity of the potential protection they provide against parasites and pathogens, and the history of individual exposure to those antigens, in relation to the environmental where they have been living, are critical to consider. This is especially the case for maternal antibodies because the exposure to antigens leading to an immune response of the females that transfer those antibodies will occur at a different time and location compared to when and where those antibodies may play a role in the offspring. Some theoretical work has outlined such issues (Garnier et al. 2012) and it would be important to stress this more clearly in the Introduction and/or when discussing issues about possible cross reactivity of the antibodies against the various endoparasites against which the sheep are exposed.

Cited references not cited in the text:

Garnier, R., Boulinier, T. & Gandon, S. 2012. Coevolution between maternal transfer of immunity and other resistance strategies against pathogens. *Evolution* 66-10: 3067-3078.

Gomez-Chamorro A, Heinrich V, Sarr A, Roethlisberger, O., Genné, D., Bregnard, C., Jacquet, M., & Voordouw, M. J. 2019. Maternal antibodies provide bank voles with strain-specific protection against infection by the Lyme disease pathogen. *Applied and Environmental Microbiology* 85 : e01887-19.

Staszewski, V., Reece, S.E., O'Donnell, A.J. & Cunningham, E.J.A. 2012. Drug treatment of malaria infections can reduce levels of protection transferred to offspring via maternal immunity. *Proc. R. Soc. B.* 279 : 2487-2496.

Other remarks:

Lines 105-143: Very well written methodological part about the data and samples used: the information are well put into the context of the detailed previous knowledge acquired on the system.

Lines 126 and 137 : What are isotypes ? Please define what they are as not all readers will know. Also, are IgA, IgG and IgE expected to play various roles and/or to be transmitted differently from the mother to the lamb? Were the different distributions of antibody levels provided on Figure S2 expected? (with many lambs with small OD values of anti-Tc IgA, but a much more normal distribution for the anti-Tc IgG) What could they mean? Elements about the quantification of the plasma levels of the different isotypes are well described in the supplement part of the manuscript, but it could be important to outline early in the manuscript some general and possibly specific elements regarding what is known of those isotypes in relation to the topic of the study. General elements provided by e.g. Tizard 2012 could for instance be useful in the Introduction or Methods. It could for instance be important to outline whether and how the levels of the different isotypes were expected to be correlated ?

Line 153: What is known of the levels of maternal antibodies? Are they repeatable? Do they vary in space within the sampled population? As those levels should directly influence those of the lambs, and the study is correlative, this is an important issue, as potential correlated factors, such the history of exposure to the parasites and the level of the female immune response, may relates to space use by mothers or other individuals characteristics related to their immune response. I

see that it is explained at the end of the paragraph (lines 164-166) that the female antibody level is considered at the last step of the analyses, but not much elements are provided about this. I guess this is because this analysis is only done on the subset of lamb/mothers with data available on the mother, but it would be important to outline why the mother antibody is considered, and whether antibody level in August one year is likely to reflect antibody level at the time of birth.

Line 189: Why now considering IgA as an explanation variable for IgG? What is the underlying biological hypothesis? Ok, elements are provided on lines 192-193, but it would be better to outline some of this before (even in the Introduction). Ok, sorry this was IgA in August not just after birth. But then the fact that the result is the same as when August IgG is considered is stated without saying whether this was expected or not (lines 194-195). The inference is much stronger when expected relationships are explored, and thus it is good if you could justify more clearly why you tested such relationships.

Lines 233-235: 'There was a quadratic association of maternal age with neonatal anti-Tc IgE and IgG and total IgG levels, in which offspring of the youngest and eldest mothers had lower antibody levels'. Was this predicted? What is the hypothesis behind? Is this linked to exposure? Are specific IgG levels expected to last a long time following exposure to those endoparasites? Again, the inference is much stronger when expected relationships are explored, and the message to the reader would be much clearer.

Line 238: Please replace 'antibodies' by 'antibody levels' as it is not antibodies that are correlated (especially as total IgG and anti-Tc IgG are considered).

Lines 251-252: Why testing each antibody levels together or independently? What is the underlying hypothesis?

Lines 296-298: 'Maternal identity explained a considerable proportion of observed variation in offspring neonatal parasite-specific antibody levels, showing that females are consistent in levels of antibodies transferred to offspring': is this mostly expected to be due to differences in their propensity to transfer antibodies, or due to their consistent inter-year level of antibodies? (see Bouludier & Staszewski 2008 for a discussion on this important issue).

Lines 300-301: What about the role of the history of parasite exposure of females? Are all females expected to be exposed in the same way in the study population?

Lines 301-303: 'Importantly, we have been able to demonstrate that among-female variation in maternal transfer of antibodies is under natural selection via both offspring neonatal and first winter survival': But is there any genetic basis for 'among-female variation in maternal transfer of antibodies'? This would also be important in terms of the evolution of the maternal transfer of antibodies and its implications (See Grinstaff et al. 2003 and Bouludier & Staszewski 2008 for discussion). This could be mentioned as a perspective to explore.

Lines 320-322: 'it seems likely that the effect of maternal age on maternal transfer of antibodies in Soay sheep reflects reduced condition or immune development and function in the youngest and eldest females'. Any suggestion of how this could be explored experimentally?

Line 347: "suggests a potential role for protective maternal transfer of immunity." The need for experimental approaches as perspectives could be stressed there (or see below).

Lines 349-350: "mothers continued to deliver antibodies to the offspring's gut via colostrum and milk long after the gut closed, and these maternal antibodies have a role in local protection in the gut from infection during lactation": why is this not outlined in the Introduction ?

Line 357: As it was done in rodents in less natural settings (e.g., Staszewski et al. 2012, Gomez-Chamorro et al. 2019), experimental approaches would also be important to confirm the causal nature of some of the key reported relationships. Such studies have also addressed the key issue of the specificity of the protective effects of the maternal antibodies.

Decision letter (RSPB-2020-1931.R0)

02-Oct-2020

Dear Dr Sparks

I am pleased to inform you that your manuscript RSPB-2020-1931 entitled "Maternally-derived anti-helminth antibodies predict offspring survival in a wild mammal" has been accepted for publication in Proceedings B.

The referee(s) have recommended publication, but also suggest some minor revisions to your manuscript. Therefore, I invite you to respond to the referee(s)' comments and revise your manuscript. Because the schedule for publication is very tight, it is a condition of publication that you submit the revised version of your manuscript within 7 days. If you do not think you will be able to meet this date please let us know.

It is a condition of publication that data supporting your paper are made available either in the electronic supplementary material or through an appropriate repository. Please see our Data Sharing Policies <https://royalsociety.org/journals/authors/author-guidelines/#data>.

In order to ensure effective and robust dissemination and appropriate credit to authors the dataset(s) used should be fully cited. To ensure archived data are available to readers, authors

should include a 'data accessibility' section immediately after the acknowledgements section. This should list the database and accession number for all data from the article that has been made publicly available, for instance:

[http://datadryad.org/submit?journalID=RSPB&manu=\(Document not available\)](http://datadryad.org/submit?journalID=RSPB&manu=(Document%20not%20available)) which will take you to your unique entry in the Dryad repository. If you have already submitted your data to dryad you can make any necessary revisions to your dataset by following the above link. Please see <https://royalsociety.org/journals/ethics-policies/data-sharing-mining/> for more details.

Sincerely,

Dr Sasha Dall

Associate Editor

Board Member: 1

Comments to Author:

Two expert reviewers have provided their assessment of the manuscript. Their reviews are very clear and constructive and I agree with all of their detailed suggestions. Reviewer 1 in gives advice on improving the clarity of writing, particularly in terms of explaining the motivation for the study. They request clarification on the timing of analysis for antibody levels relative to the time the samples were taken, and suggest analysis could be done to control for potential effects of sample degradation. Reviewer 2 highlights some relevant theoretical work which could be integrated, and suggests the authors discuss the limitation in being an entirely observational study and give more reference to experimental papers on the topic. Finally, both reviewers raise the concern that the different immunoglobulins (IgG, IgE, IgA) require more explanation in terms of the mechanisms and the biological justification for comparing them. I agree that this could help increase clarity in the manuscript, in particular for the broader readership of Proceedings B.

As for my own reading of this manuscript, I found this to be a compelling study providing novel insights for the field of adaptive maternal effects by showing how maternal antibody transfer confers fitness benefits to offspring in a free-living vertebrate population. The manuscript is well written and figures are generally clear. My only minor comment beyond the helpful suggestions raised by the reviewers is that Figure 1 could be made more visually appealing and less cluttered, using smaller arrow heads, and boxes shaded rather than with outlines to avoid all the thick lines. I would also explain in the figure legend why there are some relationships in (a) that do not appear in (b) and vice versa, even though this is explained in the methods, as a reader skimming the paper may otherwise think there has been some mistake. I also note that the data were not provided at the point of submission, so please can these be made available following the instructions at <https://royalsociety.org/journals/authors/author-guidelines/#data>.

Reviewer(s)' Comments to Author:

Referee: 1

Comments to the Author(s)

In this manuscript, the authors utilize a 25-year dataset from Soay sheep naturally infected with a strongyle parasite to test if maternally-derived antibodies impact later parasite burdens, offspring size and first winter survival. The authors measured parasite-specific levels of IgG, IgE and IgA, and total IgG levels. Total IgG levels predicted offspring growth and was interpreted as a measure of nutritional contributions from mothers to offspring. They found a positive association between levels of all four antibody measures and survival to four months; however, parasite-specific IgG levels predicted lamb survival independently. Furthermore, this relationship was independent of lamb body weight. There was also an indirect effect of parasite-specific IgG levels on lamb survival. Lambs with higher parasite-specific IgG shortly after birth had lower fecal egg counts of the strongyle parasite in the summer and lower fecal egg counts predicted higher survival. This is a compelling demonstration from a natural population that variation among females in maternal antibody transfer can have important consequences for offspring parasite resistance, growth, and survival.

General Comments:

1. The critical statements in the Introduction and Discussion that should frame what the authors wanted to learn from the study should be re-written to increase clarity and impact by following the structure "to learn x, we did y". For example, the last paragraph of the Introduction beginning at line 88 (and most of this paragraph) is a statement of "we did y" but does not clearly communicate what the authors wanted to learn or what knowledge gap they wanted to fill by using these samples and analyses. Similarly, the first paragraph of the Discussion beginning at line 295 is a statement of what was done, rather than what was learned. In both instances, the critical information you want to communicate about what you hoped to learn from the study is buried later in the paragraphs. These paragraphs would be more impactful if the statements of what you wanted to learn were placed at the beginning before describing the techniques.
2. When were the plasma samples analysed to measure antibody levels relative to sample collection? Were all of the samples assayed at one time or were they assayed as they were collected? If the samples were all analysed at the same time, then antibody levels from the oldest samples would be expected to be lower simply due to sample degradation over the 25-year period. It is not entirely clear but it seems all samples were assayed at the same time based on this sentence (line 26-27 supplementary info), "In order to minimise confounding capture year effects with plate to plate variation, each plate included samples from two years paired at random." It is critical to account for potential effects of antibody degradation in storage due to sample age or at least demonstrate that sample storage time was unrelated to the measured antibody levels.
3. Given that IgA is primarily associated with the gut and IgE is the primary mediator of the parasitic immune response, it was surprising that parasite-specific IgG levels were the strongest predictor of lamb survival in the first year. Through what potential mechanisms do you think IgG provides protection in this system that would be unique and distinct from IgA and IgE?

Specific Comments:

1. The Abstract could be strengthened to better frame the knowledge gap. For example, the first sentence tells readers that maternal antibodies are crucial for survival and then, the second sentence tells readers that we don't really know if maternal antibodies are critical for fitness.
2. Line 107: To people unfamiliar with this population, individuals living in the Village might be assumed to refer to human individuals.
3. Line 126: What do you mean by "IgG, IgA and IgE activity against antigens"?
4. Lines 314-315: A reminder of how maternal condition was quantified would be helpful.

Referee: 2

Comments to the Author(s)

This manuscript presents a well-conducted correlative study on an important topic, the potential effect of maternal antibodies in natural settings. It is based on a unique data set and the findings are clear, highlighting the potential role of maternally transferred immunoglobulins on the

protection of lambs against naturally occurring endoparasites to which they are exposed after birth in a wild living sheep population.

In general, the manuscript is well written, the analyses are sound and the figures and tables provide useful results. It has nevertheless some weaknesses that require some consideration:

(1) The fact that the study is not experimental is a weakness that needs to be acknowledged, and it is important that readers are made aware that earlier experimental studies have been carried out on the topic, even if for most of those it was done using relatively artificial settings involving captive rodents (e.g., Staszewski et al. 2012, Gomez-Chamoro et al. 2019).

(2) The potential role of the different types of immunoglobulin isotypes, IgA, IgE and IgG, is not discussed and specifically investigated. Even if not much information is available from the literature, could this be outlined more clearly? (see other comments below)

(3) When considering the role of antibodies in epidemiology, ecology and the evolution of immune defenses, the specificity of the potential protection they provide against parasites and pathogens, and the history of individual exposure to those antigens, in relation to the environmental where they have been living, are critical to consider. This is especially the case for maternal antibodies because the exposure to antigens leading to an immune response of the females that transfer those antibodies will occur at a different time and location compared to when and where those antibodies may play a role in the offspring. Some theoretical work has outlined such issues (Garnier et al. 2012) and it would be important to stress this more clearly in the Introduction and/or when discussing issues about possible cross reactivity of the antibodies against the various endoparasites against which the sheep are exposed.

Cited references not cited in the text:

Garnier, R., Boulinier, T. & Gandon, S. 2012. Coevolution between maternal transfer of immunity and other resistance strategies against pathogens. *Evolution* 66-10: 3067–3078.

Gomez-Chamorro A, Heinrich V, Sarr A, Roethlisberger, O., Genné, D., Bregnard, C., Jacquet, M., & Voordouw, M. J. 2019. Maternal antibodies provide bank voles with strain-specific protection against infection by the Lyme disease pathogen. *Applied and Environmental Microbiology* 85 : e01887-19.

Staszewski, V., Reece, S.E., O'Donnell, A.J. & Cunningham, E.J.A. 2012. Drug treatment of malaria infections can reduce levels of protection transferred to offspring via maternal immunity. *Proc. R. Soc. B.* 279 : 2487–2496.

Other remarks:

Lines 105-143: Very well written methodological part about the data and samples used: the information are well put into the context of the detailed previous knowledge acquired on the system.

Lines 126 and 137 : What are isotypes ? Please define what they are as not all readers will know. Also, are IgA, IgG and IgE expected to play various roles and/or to be transmitted differently from the mother to the lamb? Were the different distributions of antibody levels provided on Figure S2 expected? (with many lambs with small OD values of anti-Tc IgA, but a much more normal distribution for the anti-Tc IgG) What could they mean? Elements about the quantification of the plasma levels of the different isotypes are well described in the supplement part of the manuscript, but it could be important to outline early in the manuscript some general and possibly specific elements regarding what is known of those isotypes in relation to the topic of the study. General elements provided by e.g. Tizard 2012 could for instance be useful in the Introduction or Methods. It could for instance be important to outline whether and how the levels of the different isotypes were expected to be correlated ?

Line 153: What is known of the levels of maternal antibodies? Are they repeatable? Do they vary in space within the sampled population? As those levels should directly influence those of the lambs, and the study is correlative, this is an important issue, as potential correlated factors, such the history of exposure to the parasites and the level of the female immune response, may relates to space use by mothers or other individuals characteristics related to their immune response. I see that it is explained at the end of the paragraph (lines 164-166) that the female antibody level is considered at the last step of the analyses, but not much elements are provided about this. I guess this is because this analysis is only done on the subset of lamb/mothers with data available on the mother, but it would be important to outline why the mother antibody is considered, and whether antibody level in august one year is likely to reflect antibody level at the time of birth.

Line 189: Why now considering IgA as an explanation variable for IgG? What is the underlying biological hypothesis? Ok, elements are provided on lines 192-193, but it would be better to outline some of this before (even in the Introduction). Ok, sorry this was IgA in August not just after birth. But then the fact that the result is the same as when August IgG is considered is stated without saying whether this was expected or not (lines 194-195). The inference is much stronger when expected relationships are explored, and thus it is good if you could justify more clearly why you tested such relationships.

Lines 233-235: 'There was a quadratic association of maternal age with neonatal anti-Tc IgE and IgG and total IgG levels, in which offspring of the youngest and eldest mothers had lower antibody levels'. Was this predicted? What is the hypothesis behind? Is this linked to exposure? Are specific IgG levels expected to last a long time following exposure to those endoparasites? Again, the inference is much stronger when expected relationships are explored, and the message to the reader would be much clearer.

Line 238: Please replace 'antibodies' by 'antibody levels' as it is not antibodies that are correlated (especially as total IgG and anti-Tc IgG are considered).

Lines 251-252: Why testing each antibody levels together or independently? What is the underlying hypothesis?

Lines 296-298: 'Maternal identity explained a considerable proportion of observed variation in offspring neonatal parasite-specific antibody levels, showing that females are consistent in levels of antibodies transferred to offspring': is this mostly expected to be due to differences in their propensity to transfer antibodies, or due to their consistent inter-year level of antibodies? (see Bouludier & Staszewski 2008 for a discussion on this important issue).

Lines 300-301: What about the role of the history of parasite exposure of females? Are all females expected to be exposed in the same way in the study population?

Lines 301-303: 'Importantly, we have been able to demonstrate that among-female variation in maternal transfer of antibodies is under natural selection via both offspring neonatal and first winter survival': But is there any genetic basis for 'among-female variation in maternal transfer of antibodies'? This would also be important in terms of the evolution of the maternal transfer of antibodies and its implications (See Grinstaff et al. 2003 and Bouludier & Staszewski 2008 for discussion). This could be mentioned as a perspective to explore.

Lines 320-322: 'it seems likely that the effect of maternal age on maternal transfer of antibodies in Soay sheep reflects reduced condition or immune development and function in the youngest and eldest females'. Any suggestion of how this could be explored experimentally?

Lines 347: "suggests a potential role for protective maternal transfer of immunity." The need for experimental approaches as perspectives could be stressed there (or see below).

Lines 349-350: "mothers continued to deliver antibodies to the offspring's gut via colostrum and milk long after the gut closed, and these maternal antibodies have a role in local protection in the gut from infection during lactation": why is this not outlined in the Introduction ?

Line 357: As it was done in rodents in less natural settings (e.g., Staszewski et al. 2012, Gomez-Chamoro et al. 2019), experimental approaches would also be important to confirm the causal nature of some of the key reported relationships. Such studies have also addressed the key issue of the specificity of the protective effects of the maternal antibodies.

Author's Response to Decision Letter for (RSPB-2020-1931.R0)

See Appendix A.

Decision letter (RSPB-2020-1931.R1)

04-Nov-2020

Dear Dr Sparks

I am pleased to inform you that your manuscript entitled "Maternally-derived anti-helminth antibodies predict offspring survival in a wild mammal" has been accepted for publication in Proceedings B.

Open Access

Paper charges

Sincerely,

Appendix A

4th November 2020

Dear Dr Dall,

Please find attached a revised and track-changes version of our manuscript “Maternally-derived anti-helminth antibodies predict offspring survival in a wild mammal” for consideration for publication in *Proceedings of the Royal Society B*. We would like to thank the editors and the reviewers for their constructive and positive comments which have much improved our manuscript. Please find below a detailed description of our responses and alterations in light of these comments, with line numbers corresponding to the track-changes version of our manuscript. We hope that our changes have made this manuscript acceptable for publication in *Proceedings of the Royal Society B*.

Best wishes,

Alex Sparks (on behalf of all co-authors)

Associate Editor

Board Member: 1

Comments to Author:

- **Two expert reviewers have provided their assessment of the manuscript. Their reviews are very clear and constructive and I agree with all of their detailed suggestions. Reviewer 1 in gives advice on improving the clarity of writing, particularly in terms of explaining the motivation for the study. They request clarification on the timing of analysis for antibody levels relative to the time the samples were taken, and suggest analysis could be done to control for potential effects of sample degradation. Reviewer 2 highlights some relevant theoretical work which could be integrated, and suggests the authors discuss the limitation in being an entirely observational study and give more reference to experimental papers on the topic. Finally, both reviewers raise the concern that the different immunoglobulins (IgG, IgE, IgA) require more explanation in terms of the mechanisms and the biological justification for comparing them. I agree that this could help increase clarity in the manuscript, in particular for the broader readership of Proceedings B.**
- **As for my own reading of this manuscript, I found this to be a compelling study providing novel insights for the field of adaptive maternal effects by showing how maternal antibody transfer confers fitness benefits to offspring in a free-living vertebrate population. The manuscript is well written and figures are generally clear. My only minor comment beyond the helpful suggestions raised by the reviewers is that Figure 1 could be made more visually appealing and less cluttered, using smaller arrow heads, and boxes shaded rather than with outlines to avoid all the thick lines. I would also explain in the figure legend why there are some relationships in (a) that do not appear in (b) and vice versa, even though this is explained in the methods, as a reader skimming the paper may otherwise think there has been some mistake.**

Thank you for the suggestion. We have amended the figure and legend and believe Figure 1 is now significantly clearer. We have also moved original Figure 2 to the supplement (now Figure S5) in order that the revised manuscript fits within 10 pages.

- **I also note that the data were not provided at the point of submission, so please can these be made available following the instructions at <https://royalsociety.org/journals/authors/author-guidelines/#data>.**

The data has been submitted to dryad (doi:10.5061/dryad.ttdz08kvx) but is set to private for the review stage until article acceptance. It can be accessed using this link -

https://datadryad.org/stash/share/D1-K84EFii8uXnLiplhZG8EcaXjzZD_uGC15oV3E8Gg

We have added a data accessibility statement and a reference for the dryad dataset to the manuscript on lines 683-684:

“DATA ACCESSIBILITY

Data are available from the Dryad Digital Repository: doi:10.5061/dryad.ttdz08kvx [62].”

- **Reviewer(s)' Comments to Author:**

Referee: 1

Comments to the Author(s)

- **In this manuscript, the authors utilize a 25-year dataset from Soay sheep naturally infected with a strongyle parasite to test if maternally-derived antibodies impact later parasite burdens, offspring size and first winter survival. The authors measured parasite-specific levels of IgG, IgE and IgA, and total IgG levels. Total IgG levels predicted offspring growth and was interpreted as a measure of nutritional contributions from mothers to offspring. They found a positive association between levels of all four antibody measures and survival to four months; however, parasite-specific IgG levels predicted lamb survival independently. Furthermore, this relationship was independent of lamb body weight. There was also an indirect effect of parasite-specific IgG levels on lamb survival. Lambs with higher parasite-specific IgG shortly after birth had lower fecal egg counts of the strongyle parasite in the summer and lower fecal egg counts predicted higher survival. This is a compelling demonstration from a natural population that variation among females in maternal antibody transfer can have important consequences for offspring parasite resistance, growth, and survival.**

General Comments:

- **1. The critical statements in the Introduction and Discussion that should frame what the authors wanted to learn from the study should be re-written to increase clarity and impact by following the structure “to learn x, we did y”. For example, the last paragraph of the Introduction beginning at line 88 (and most of this paragraph) is a statement of “we did y” but does not clearly communicate what the authors wanted to learn or what knowledge gap they wanted to fill by using these samples and analyses. Similarly, the first paragraph of the Discussion beginning at line 295 is a statement of what was done, rather than what was learned. In both instances, the critical information you want to communicate about what you hoped to learn from the study is buried later in the paragraphs. These paragraphs would be more impactful if the statements of what you wanted to learn were placed at the beginning before describing the techniques.**

We have rewritten the final paragraph of the introduction (lines 111-152) and the start of the discussion (lines 414-453) following the reviewer's suggestion.

- **2. When were the plasma samples analysed to measure antibody levels relative to sample collection? Were all of the samples assayed at one time or were they assayed as they were collected? If the samples were all analysed at the same time, then antibody levels from the oldest samples would be expected to be lower simply due to sample degradation over the 25-year period. It is not entirely clear but it seems all samples were assayed at the same time based on this sentence (line 26-27 supplementary info), “In order to minimise confounding capture year effects with plate to plate variation, each plate included samples from two years paired at random.” It is critical to account for potential effects of antibody degradation in storage due to sample age or at least demonstrate that sample storage time was unrelated to the measured antibody levels.**

The plasma samples were all assayed together between 13/5/2016-26/7/2016. We investigated an association between sample year and antibody levels and found no evidence for a positive association between sample year and antibody levels of any of the four measures indicative of sample degradation. We actually find significant but very low and negative associations between sample year and antibody levels but with large variation between years (Figure S2). We have added the boxplot to the supplement (Figure S2), a line to the methods (205-206) and full details to the supplementary methods due to limitations on manuscript length (lines 66-73):

“There was no evidence of sample degradation in our dataset (see Supplementary Methods, Figure S2).”

“All samples were assayed between 13/5/2016-26/7/2016 but were collected over many years and stored at -20°C until they were assayed. We checked for possible signatures of sample degradation over time since collection but we found no positive temporal trend in levels of any of the four antibody measures as expected with sample degradation. In fact we found evidence for negative, but weak, temporal trends in the levels of the four antibody measures with sample year (IgA: $\beta = -0.005 \pm 0.001$ SE, $F_1 = 13.540$, $p < 0.001$; IgE: $\beta = -0.015 \pm 0.002$ SE, $F_1 = 68.811$, $p < 0.001$; IgG: $\beta = -0.002 \pm 0.001$ SE, $F_1 = 9.182$, $p = 0.002$; total IgG: $\beta = -0.001 \pm 0.001$ SE, $F_1 = 3.915$, $p = 0.048$), however, there was considerable variation between years (Figure S2) suggesting sample degradation was not present.”

- **3. Given that IgA is primarily associated with the gut and IgE is the primary mediator of the parasitic immune response, it was surprising that parasite-specific IgG levels were the strongest predictor of lamb survival in the first year. Through what potential mechanisms do you think IgG provides protection in this system that would be unique and distinct from IgA and IgE?**

We have added in a paragraph to the discussion to discuss the role of different isotypes in helminth infections (lines 629-646):

“Previous studies on protective immunity to *T. circumcincta* infection have mainly focused on the importance of IgA and IgE. In lambs, reduced FEC is associated with increased IgA levels directed at worm growth, fecundity, and the inhibition of larvae development [46,47]. In resistant sheep, a hypersensitivity response results in the expulsion of incoming larvae, and the arrestment of larvae development moderated partly by IgA [48]. IgE levels and IgE-dependent mast cell degranulation are also negatively associated with FEC and worm burden [49,50]. While IgA and IgE are produced more locally at the mucosa, IgG is the dominant antibody isotype in circulation [15]. Studies investigating the role of IgG in *T. circumcincta* infection have documented negative associations between parasite-specific IgG and FEC or worm burden [51,52], while IgG has been implicated indirectly by protective effects of complement in other ruminant-worm interactions [53,54]. In laboratory mice, IgG has been documented to have an important role in passive immunity, and mediating protective immunity, to several nematode species [55–58]. Previously, using samples collected in August, we found that circulating levels of anti-Tc IgG, but not IgA or IgE, predict over-winter survival in adults, but there was no association between any of these antibodies and over-winter survival in lambs [22]. We have also shown that high plasma anti-Tc IgG is associated with high local anti-Tc IgG and IgA [59]. Our results provide further evidence of the importance of the IgG isotype in resistance to strongyle infection in this system, via associations between maternally-derived levels of anti-Tc IgG and offspring survival.”

Specific Comments:

- **1. The Abstract could be strengthened to better frame the knowledge gap. For example, the first sentence tells readers that maternal antibodies are crucial for survival and then, the second sentence tells readers that we don't really know if maternal antibodies are critical for fitness.**

We have clarified that much is known in lab/human/domestic animals, but very little is known in the wild. We have updated line 19 of the abstract to read:

“The transfer of antibodies from mother to offspring provides crucial protection against infection to offspring during early life in humans and domestic and laboratory animals.

However, few studies have tested the consequences of variation in maternal antibody transfer for offspring fitness in the wild.”

- **2. Line 107: To people unfamiliar with this population, individuals living in the Village might be assumed to refer to human individuals.**

We have replaced individuals with sheep on line 158:

“Sheep living in the Village Bay area of the island of Hirta have been the subject of a long-term study since 1985 [23].”

- **3. Line 126: What do you mean by “IgG, IgA and IgE activity against antigens”?**

We have updated this sentence on line 176:

“Levels of IgA, IgG and IgE antibodies binding to antigens of the third larval stage of *T. circumcincta*”

- **4. Lines 314-315: A reminder of how maternal condition was quantified would be helpful.**

We have clarified this and updated the sentence on lines 455-456 to read:

“Consistent with previous studies, neonatal antibody levels were related to maternal age and the mother’s own previous immune phenotype.”

Referee: 2

- **Comments to the Author(s)**
- **This manuscript presents a well-conducted correlative study on an important topic, the potential effect of maternal antibodies in natural settings. It is based on a unique data set and the findings are clear, highlighting the potential role of maternally transferred immunoglobulins on the protection of lambs against naturally occurring endoparasites to which they are exposed after birth in a wild living sheep population.**
- **In general, the manuscript is well written, the analyses are sound and the figures and tables provide useful results. It has nevertheless some weaknesses that require some consideration:**
 - **(1) The fact that the study is not experimental is a weakness that needs to be acknowledged, and it is important that readers are made aware that earlier experimental studies have been carried out on the topic, even if for most of those it was done using relatively artificial settings involving captive rodents (e.g., Staszewski et al. 2012, Gomez-Chamoro et al. 2019).**

We do highlight some limitations of observational approaches in our Introduction (lines 75-94). However, observation of naturally-occurring associations between phenotypes and fitness is the only way we can actually quantify natural selection as it truly operates in the wild. Experiments cannot achieve this. This is a key strength of our approach, which the reviewer appears to appreciate. To clarify this to readers, we have added the following (lines 72-75):

“Measuring associations between phenotypes and fitness under natural conditions is by necessity observational rather than experimental. However, this provides insight into the operation of natural selection which cannot be obtained through experimental approaches.”

- **(2) The potential role of the different types of immunoglobulin isotypes, IgA, IgE and IgG, is not discussed and specifically investigated. Even if not much information is available from the literature, could this be outlined more clearly? (see other comments below)**

We have added a paragraph to the supplementary methods (due to limitations on manuscript length) about the different immunoglobulin isotypes (lines 75-94, see response below) and a paragraph to the discussion about the role of the different isotypes in helminth infections in domestic ruminants and mice (lines 629-646, see response to Reviewer 1 above).

- **(3) When considering the role of antibodies in epidemiology, ecology and the evolution of immune defenses, the specificity of the potential protection they**

provide against parasites and pathogens, and the history of individual exposure to those antigens, in relation to the environmental where they have been living, are critical to consider. This is especially the case for maternal antibodies because the exposure to antigens leading to an immune response of the females that transfer those antibodies will occur at a different time and location compared to when and where those antibodies may play a role in the offspring. Some theoretical work has outlined such issues (Garnier et al. 2012) and it would be important to stress this more clearly in the Introduction and/or when discussing issues about possible cross reactivity of the antibodies against the various endoparasites against which the sheep are exposed.

We agree, although this is a bit beyond the scope of our study. We have added a few sentences to our discussion to highlight the importance of this for future studies (lines 472-524):

“The substantial, consistent differences among mothers in neonatal antibody levels of their offspring (indicated by the large maternal effect) suggest maternal genotype and prior maternal environment may play an important role in driving the variation in maternal antibody transfer we observe in wild Soay sheep. Further study using available genetic and environmental data for this system could help elucidate their relative importance.”

- **Other remarks:**
- **Lines 105-143: Very well written methodological part about the data and samples used: the information are well put into the context of the detailed previous knowledge acquired on the system.**

Thank you.

- **Lines 126 and 137 : What are isotypes ? Please define what they are as not all readers will know. Also, are IgA, IgG and IgE expected to play various roles and/or to be transmitted differently from the mother to the lamb? Were the different distributions of antibody levels provided on Figure S2 expected? (with many lambs with small OD values of anti-Tc IgA, but a much more normal distribution for the anti-Tc IgG) What could they mean? Elements about the quantification of the plasma levels of the different isotypes are well described in the supplement part of the manuscript, but it could be important to outline early in the manuscript some general and possibly specific elements regarding what is known of those isotypes in relation to the topic of the study. General elements provided by e.g. Tizard 2012 could for instance be useful in the Introduction or Methods. It could for instance be important to outline whether and how the levels of the different isotypes were expected to be correlated ?**

We have added a sentence to the methods (lines 204-205) and a section to the supplementary methods (due to limitations on manuscript length) describing the five antibody isotypes found in mammals, their roles and their composition in colostrum and milk on lines 75-94:

“A description of the different antibody classes (isotypes) in mammals is available in the Supplementary Methods.”

“Mammals have five classes (isotypes) of antibodies: IgA, IgE, IgD, IgG and IgM. IgG is the most abundant isotype in serum, and is effective at neutralising toxins, activating complement and the agglutination and opsonisation of pathogens. IgM is the second-commonest isotype in serum, and is the major isotype produced following first exposure to an antigen. IgM forms pentamers making them effective activators of the complement system but prevents them readily leaving the bloodstream. The third most common isotype in serum is IgA, which is the most common isotype in secretions including mucosal surfaces and it acts as a strong neutralising antibody and can agglutinate pathogens. IgD can be found on B cells and this isotype tends not to be secreted. Finally, IgE is present at very low concentrations in serum, has the lowest half-life of all antibodies, but is key in mediating acute inflammatory responses. Colostrum is rich in immunoglobulins, predominantly IgG in ewes, followed by IgM and IgA. In milk, the concentration of immunoglobulins drops and IgG continues to predominate but IgA becomes the second most dominant isotype [1]. In peri-parturient ewes in this population we found uniformly positive correlations between anti-Tc IgG, IgA, IgM and total IgG, IgA and IgM, suggesting that maternally-transferred antibody levels are likely to be positively correlated in neonates [2].”

- **Line 153: What is known of the levels of maternal antibodies? Are they repeatable? Do they vary in space within the sampled population? As those levels should directly influence those of the lambs, and the study is correlative, this is an important issue, as potential correlated factors, such the history of exposure to the parasites and the level of the female immune response, may relates to space use by mothers or other individuals characteristics related to their immune response. I see that it is explained at the end of the paragraph (lines 164-166) that the female antibody level is considered at the last step of the analyses, but not much elements are provided about this. I guess this is because this analysis is only done on the subset of lamb/mothers with data available on the mother, but it would be important to outline why the mother antibody is considered, and whether antibody level in august one year is likely to reflect antibody level at the time of birth**

We know that anti-*T. circumcincta* IgA, IgE and IgG levels are all very repeatable in adult Soay sheep, with high correlations between measures in consecutive years (Sparks et al., 2019). While this previous study looked at plasma levels of antibodies at one time point in the year, it does suggest there is some stability in anti-helminth immune responses in this

population. How such antibody levels vary in space in the study area is currently unknown, but an important area for further study. We have added some explanation to the methods on lines 259-263:

“In ruminant colostrum, the majority of immunoglobulins are derived by transfer from the bloodstream [15]. Previously it was shown that plasma anti-Tc IgA, IgE and IgG levels measured in August over 26 years are very repeatable in adult Soay sheep, with high correlations between measures in consecutive years [34]. As a result we predicted that antibody levels in the previous August may reflect those in April of the following year”

- **Line 189: Why now considering IgA as an explanation variable for IgG? What is the underlying biological hypothesis? Ok, elements are provided on lines 192-193, but it would be better to outline some of this before (even in the Introduction). Ok, sorry this was IgA in August not just after birth. But then the fact that the result is the same as when August IgG is considered is stated without saying whether this was expected or not (lines 194-195). The inference is much stronger when expected relationships are explored, and thus it is good if you could justify more clearly why you tested such relationships.**

We have added August before strongyle FEC and anti-Tc IgA on line 286 to make it clearer. We have then updated the text (lines 289-297) to read:

“LMMs found a negative association between neonatal anti-Tc IgG and both August anti-Tc IgA and IgG levels (Table S5). Previous work showed that lamb August anti-Tc IgA was the best predictor of August FEC [22], so we included this measure in our SEMs to capture lamb immune resistance to worm infection at weaning. We also ran the SEM with August anti-Tc IgG instead and this produced similar results (see Figure S8a-b) which is unsurprising given that these antibodies are positively correlated [22].”

- **Lines 233-235: ‘There was a quadratic association of maternal age with neonatal anti-Tc IgE and IgG and total IgG levels, in which offspring of the youngest and eldest mothers had lower antibody levels’. Was this predicted? What is the hypothesis behind? Is this linked to exposure? Are specific IgG levels expected to last a long time following exposure to those endoparasites? Again, the inference is much stronger when expected relationships are explored, and the message to the reader would be much clearer.**

We have added to the introduction (lines 67-69):

“These factors are likely to contribute to age-specific variation in maternal antibody transfer, with younger and older mothers generally having lower nutritional status and less effective immunity than prime-age mothers.”

We have added that this quadratic age effect was as expected in the discussion, and the following sentences discuss the maternal age effect (lines 456-463):

“As expected, we found that neonatal antibody levels tended to be lower in offspring of the youngest and eldest mothers. Such quadratic maternal age effects are widely observed in wild vertebrates [38–40], and the pattern mirrors trends with age in survival, reproductive performance, weight and parasite burden in the Soay sheep [41]. Maternal age is a strong and widespread predictor of maternal condition and performance in studies of wild vertebrates [42], and it seems likely that the effect of maternal age on maternal antibody transfer in Soay sheep reflects reduced condition or immune development and function in the youngest and eldest females.”

- **Line 238: Please replace ‘antibodies’ by ‘antibody levels’ as it is not antibodies that are correlated (especially as total IgG and anti-Tc IgG are considered).**

We have replaced ‘antibodies’ with ‘antibody levels’ on line 345.

- **Lines 251-252: Why testing each antibody levels together or independently? What is the underlying hypothesis?**

We initially tested all antibody levels independently to look at whether any were associated with health-related traits and survival. However, we also note that neonatal antibody levels are positively correlated with one another ($r=0.323-0.776$), so for neonatal survival where more than one antibody measure was associated with survival we also ran a model with all antibody measures in to test which antibody levels were independently associated with survival. The text now reads (lines 357-360):

“There were positive associations between all four neonatal antibody levels and survival to four months (Table S7). However, neonatal antibody levels are correlated, and so we tested independent associations between antibody levels and survival by including all four antibody measures in the same model. In this model, only neonatal anti-Tc IgG levels predicted survival independently”

- **Lines 296-298: ‘Maternal identity explained a considerable proportion of observed variation in offspring neonatal parasite-specific antibody levels, showing that females are consistent in levels of antibodies transferred to offspring’: is this mostly expected to be due to differences in their propensity to transfer antibodies, or due to their consistent inter-year level of antibodies? (see Bouludier & Staszewski 2008 for a discussion on this important issue).**

We have added that this is unknown to the discussion on lines 418-419:

“Whether this is due to consistent levels of antibodies transferred to offspring or consistent inter-year immune responses of females is unknown”

- **Lines 300-301: What about the role of the history of parasite exposure of females? Are all females expected to be exposed in the same way in the study population?**

We have added to the discussion on lines 466-472:

“Given that exposure to worms starts early, we would expect that all females would be sufficiently exposed to have developed an acquired immune response to strongyles by sexual maturity, despite exposure varying in time and space [19]. Given the high repeatability of antibody levels in adults, and positive associations with fitness and negative associations with FEC, we believe that these antibody levels reflect variation in the immune response rather than just differences in exposure in this population [22,34].”

- **Lines 301-303: ‘Importantly, we have been able to demonstrate that among-female variation in maternal transfer of antibodies is under natural selection via both offspring neonatal and first winter survival’: But is there any genetic basis for ‘among-female variation in maternal transfer of antibodies’? This would also be important in terms of the evolution of the maternal transfer of antibodies and its implications (See Grinstaff et al. 2003 and Boulinier & Staszewski 2008 for discussion). This could be mentioned as a perspective to explore.**

We agree and have added this to the discussion on lines 472-524:

“The substantial, consistent differences among mothers in neonatal antibody levels of their offspring (indicated by the large maternal effect) suggest maternal genotype and prior maternal environment may play an important role in driving the variation in maternal antibody transfer we observe in wild Soay sheep. Further study using available genetic and environmental data for this system could help elucidate their relative importance.”

- **Lines 320-322: ‘it seems likely that the effect of maternal age on maternal transfer of antibodies in Soay sheep reflects reduced condition or immune development and function in the youngest and eldest females’. Any suggestion of how this could be explored experimentally?**

This is an interesting suggestion, but as discussed above, ours is not an experimental system and the strengths of our study lie in the ability to accurately quantify naturally-occurring relationships between phenotypes and fitness components (i.e. natural selection). We do not feel discussion of such follow-on experiments are appropriate given the impossibility of such (or indeed any) experiments in our system and the limitations on manuscript length.

- **Lines 347: “suggests a potential role for protective maternal transfer of immunity.” The need for experimental approaches as perspectives could be stressed there (or see below).**

See our response below.

- **Lines 349-350: “mothers continued to deliver antibodies to the offspring’s gut via colostrum and milk long after the gut closed, and these maternal antibodies have a role in local protection in the gut from infection during lactation”:** why is this not outlined in the Introduction ?

We have added this to the introduction (lines 100-101):

“After gut closure, immunoglobulins in milk continue to provide local protection against enteric infection [15].”

- **Line 357: As it was done in rodents in less natural settings (e.g., Staszewski et al. 2012, Gomez-Chamoro et al. 2019), experimental approaches would also be important to confirm the causal nature of some of the key reported relationships. Such studies have also addressed the key issue of the specificity of the protective effects of the maternal antibodies.**

We have previously stated in the methods that these anti-*T. circumcincta* antibodies actually bind to antigens from a range of nematodes (lines 202-204, Froy et al., 2019). While we do not work on an experimental system, the temporal separation makes directionality possible to infer (method, lines 297-299). We have also added into the discussion a paragraph which emphasises the impressive work carried out on domestic animals and lab mice that have looked at the protective effect of antibodies (lines 629-646, see response to reviewer 1) as well as a sentence to the discussion highlighting the two captive rodent studies suggested (lines 465-466):

“Previous experimental studies in captive rodents have demonstrated that maternal antibodies provide strain-specific protection which depends on the infection history of the mother [43,44].”